

**A dataset of energy, water vapor and carbon exchange observations**
**in oasis-desert areas from 2012 to 2021 in a typical endorheic basin**
Shaomin Liu[1], Ziwei Xu[1], Tao Che[2], Xin Li[3], Tongren Xu[1], Zhiguo Ren[2],
Yang Zhang[2], Junlei Tan[2], Lisheng Song[4], Ji Zhou[5], Zhongli Zhu[1], Xiaofan
Yang[1], Rui Liu[6], Yanfei Ma[7]
*Correspondence to*: Shaomin Liu (smliu@bnu.edu.cn)
[1]State Key Laboratory of Earth Surface Processes and Resource Ecology, Faculty of Geographical
Science, Beijing Normal University, Beijing 100875, China
[2]Northwest Institute of Eco-Environment and Resources, Chinese Academy of Sciences, Lanzhou
730000, China
[3]National Tibetan Plateau Data Center, State Key Laboratory of Tibetan Plateau Earth System and
Resources Environment, Institute of Tibetan Plateau Research, Chinese Academy of Sciences,
Beijing 100101, China
[4]Key Laboratory of Earth Surface Processes and Regional Response in the Yangtze-Huaihe River
Basin, School of Geography and Tourism, Anhui Normal University, Wuhu 241000, China;
[5]School of Resources and Environment, University of Electronic Science and Technology of China,
Chengdu 611731, China
[6] Institute of Urban Study, School of Environmental and Geographical Sciences (SEGS), Shanghai
Normal University, Shanghai 200234, China
[7]Hebei Technology Innovation Centre for Remote Sensing Identification of Environmental Change,
Hebei Key Laboratory of Environmental Change and Ecological Construction, School of
Geographical Sciences, Hebei Normal University, Shijiazhuang 050024, China





**Abstract:**
Oases and deserts generally act as a landscape matrix and mosaic in arid/semiarid
regions. The significant difference of thermal and dynamic characteristics between
oasis and desert surface will result in oasis-desert interaction. That is, the interaction
between oasis and desert system through the exchange of momentum, energy, water
and carbon, which can lead to a series of microclimate effects that affect the structure
of the atmospheric boundary layer, changes of carbon sources/sinks in oasis and the
local ecological environment. Therefore, studying water, heat and carbon exchange is
significant for achieving the goals of carbon peaking and carbon neutrality in oasis-
desert areas and supporting the ecological security and sustainable development of
oases. To monitor energy, water vapor and carbon exchange between the land surface
and atmosphere, a land surface process integrated observatory was established in the
oasis-desert area in the middle and lower reaches of the Heihe River Basin, the 2$^{nd}$
largest endorheic basin in China. In this study, we present a suite of observational
datasets in artificial and natural oases-desert systems, which consist of long-term energy,
water vapor, carbon/methane fluxes, and auxiliary data involving hydrometeorology,
vegetation and soil parameters from 2012 to 2021. Half-hourly turbulent flux data were
acquired by an eddy covariance system and scintillometer. The hydrometeorological
data, including radiation, soil heat flux and soil temperature profile, gradient of air
temperature/humidity and wind speed/direction, air pressure, precipitation and soil
moisture profiles, were observed from automatic weather stations with a 10-minute
average period as well as the groundwater table data. Moreover, vegetation and soil

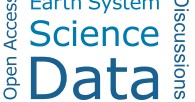

parameters were also supplemented in the datasets. Careful data processing and quality
control are implemented during data production, including data collection, processing,
archiving and sharing. The current datasets can be used to explore the water-heat-carbon
process and its influence mechanism, calibrate and validate related remote sensing
products, simulate energy, water vapor and carbon exchange in oasis and desert areas,
and provide references and representatives for other similar artificial and natural oases
along the Silk Road. The datasets are available from the National Tibetan Plateau Third
Pole Environment.

**1. Introduction**
Arid and semiarid regions represent approximately 30% of the global terrestrial surface
area (Dregne, 1991; Scanlon et al., 2006), and dryland expansion occurs under climate
change, especially in developing countries (Huang et al., 2015). This proportion is much
higher in China, as (semi)arid regions account for approximately 47% of its terrestrial
surface (Zhang et al., 2016a; Mao et al., 2018). An oasis is a unique ecological
landscape in arid and semiarid areas, which is not only the core of its ecological
environment but also the foundation of its economic development, especially in western
China, which has been an important part of the 'Silk Road' since ancient times. Oases
with less than 10% of the total area of arid regions support more than 90% of the
population in the arid regions of China (Chu et al., 2005; Li et al., 2016; Zhou et al.,
2022). The main geomorphologic feature is a wide sandy desert or Gobi (gravel desert),
interspersed with many oases of different sizes and shapes in the middle and lower



reaches of a typical endorheic basin in Northwest China (Cheng et al., 1999). The water
from upstream is the link connecting these ecosystems, and the oasis is the place where
human beings live. The oasis areas are now 3.3 times larger than those in the early
1950s in the region of northwestern China (Zhang et al., 2018). The oasis-desert system
plays a crucial role in maintaining a stable ecological environment and agricultural
productivity (Zhang and Zhao, 2015). However, inland river basins in arid and semiarid
areas are facing the crisis of ecological environment degradation, such as the drying up
of rivers and lakes, the degradation of natural vegetation, the intensification of land
desertification and the frequent occurrence of dust storms, especially in many inland
river basins westward along the Silk Road, such as the Tarim River Basin (Zhao et al.,
2013), Aral Sea Basin (Stanev et al., 2004; Crétaux et al., 2009), and Lake Urmia Basin
(Stone, 2015). Therefore, it is critical to maintain the balance between the oasis and
desert systems to achieve the goal of sustainable oasis development.
The particularity of the underlying surface in the oasis-desert area, e.g., the irrigation
cropland, riparian forest, sandy vegetation, seasonal snow and frozen soil, makes the
study of land–atmosphere interactions complex and needs comprehensive consideration
in such heterogeneous underlying surfaces. The dynamic and thermal characteristics of
the underlying surface of the oasis and the desert are significantly different, and the
oasis and desert systems interact and influence each other through momentum, energy,
water vapor and carbon exchange. Thus, the oasis-desert interaction will affect the
structure of the atmospheric boundary layer and the local ecological environment.
Additionally, under the influence of weather conditions, the oasis-desert interaction





results in the local circulation between oasis and desert and airflows form dynamic and
thermal inner boundary layer within the oasis (Cheng et al., 2014). These can lead to
the local microclimate characteristics of the oasis-desert area (Liu et al., 2020), such as
the wind shield effect and cold-wet island effect of the oasis, the humidity inversion
effect within the surrounding desert, and oasis carbon sources/sinks. These
microclimate effects play an important role in the self-sustaining and development of
oasis systems. Understanding the basic characteristics of energy, water vapor and
carbon exchange in oasis-desert ecosystems is important for achieving the goals of
carbon peaking and carbon neutrality in the oasis-desert area and supporting ecological
security and sustainable development of the oasis.
Extensive studies have investigated energy, water vapor and carbon exchange in
oasis-desert areas based on field and remote sensing observations (Taha et al., 1991;
Potchter et al., 2008; Xue et al., 2019; Wang et al., 2019; Zhou et al., 2022) and
numerical simulations (Chu et al., 2005; Meng et al., 2009; Georgescu et al., 2011; Liu
et al., 2020). Li et al. (2016) provided a complete sketch map of oasis and desert
interactions based on previous studies, including the oasis cold and wet island effect,
oasis wind shield effect (oasis effect), and air humidity inversion effect within the
surrounding desert (desert effect), which are important for the stability and
sustainability of the oasis-desert ecosystem (Liu et al., 2020). In addition, the oasis-
desert areas located in semiarid regions were found to be carbon sinks by previous
researchers (Tagesson et al., 2016; Wang et al., 2019), which can significantly affect
the carbon balance of arid regions and play an increasingly important role within the



global carbon cycle.
The Heihe River Basin (HRB), the second largest endorheic basin in China, is
characterized by artificial oases and natural oases in the middle and lower reaches,
respectively. Several experiments have been conducted in these oasis-desert areas, e.g.,
the Heihe River Basin Field Experiment (HEIFE) from 1990 to 1992 to conduct
comprehensive studies of atmosphere–land surface interactions over the Zhangye oasis
and desert area in the middle reaches of the HRB (Wang et al., 1992), the Jinta
experiment from 2005 and 2008 to focus on the energy and water exchange and the
atmospheric boundary over the Jinta oasis and desert area in the middle reaches of the
HRB (Wen et al., 2012), the oasis-desert area in the middle reaches and mountainous
area in upper reaches of watershed allied telemetry experimental research (WATER)
and the subsequent HiWATER (oasis-desert area in the middle and lower reaches and
mountainous area in upper reaches) to be a comprehensive simultaneous satellite–
airborne–ground observations eco-hydrological experiment (Li et al., 2009, 2013).
Thereafter, a multielement, multiscale, networked, and elaborate integrated observatory
network was established in the oasis-desert area in the middle and lower reaches and
mountainous area in upper reaches of the HRB since 2007 and completed in 2013 (Liu
et al., 2018). A quantitative understanding of the energy, water vapor and carbon
exchange in oasis-desert areas is crucial for recognizing the oasis-desert interactions
and is significant for protecting the ecological stability and socioeconomic development
of oases, and long-term observations are indispensable. The observations and research
findings from the oasis-desert area in the HRB will serve as references and



representatives for other similar artificial and natural oases along the Silk Road. To
achieve the aforementioned objective, observations should be continuously conducted,
and a high-quality dataset should be obtained.
In this paper, the integrated observatory network of the artificial and natural oasis-
desert areas in the middle and lower reaches in the HRB are introduced first, and the
observations characterizing the energy, water vapor and carbon exchange are detailed
explicated, which provides a 10-year dataset. Specifically, the spatial distribution and
design of the observation sites are summarized in Section 2. Section 3 describes the
data processing and quality control procedures. In Section 4, the energy, water vapor
and carbon fluxes and related auxiliary parameters are introduced in detail. The data
availability is documented in Section 5, and the conclusions are summarized in Section
6. This dataset can be used for comprehensive understanding of energy, water vapor
and carbon exchange in oasis-desert areas, and validating simulation results and remote
sensing products of energy, water vapor and carbon fluxes in oasis-desert areas.
**2.  A land surface process integrated observatory network in the oasis-desert area**
**of the HRB**
**2.1 Study area description**
The study areas are the middle and lower reaches of the HRB, which are located in
the arid regions of western China, provided by water from the typical cryosphere of the
upper reaches. The middle reaches, typical of the artificial oasis-desert system in
Zhangye City, the largest oasis in the Hexi Corridor, cover an area of 29,717 km$^2$ with
an oasis area of 5,560 km$^2$, while the lower reaches in Ejina Banner have a natural oasis-
desert system covering an area of 85,678 km$^2$ with an oasis area of 1,130 km$^2$ (Fig. 1).
Among the oases, agricultural oases can be traced to the history of more than 2000 years.
The annual average air temperature was 7.29 °C and 9.75 °C, and the annual
accumulated precipitation was 184.83 mm and 37.31 mm (1979-2018) in the middle
and lower reaches, respectively.
Eleven land surface fluxes and meteorological stations have been established in these
regions since 2012 with two superstations and eleven ordinary stations (Table 1),
specifically two oasis stations and three desert stations in the middle reaches and five
oasis stations and one desert station in the lower reaches.

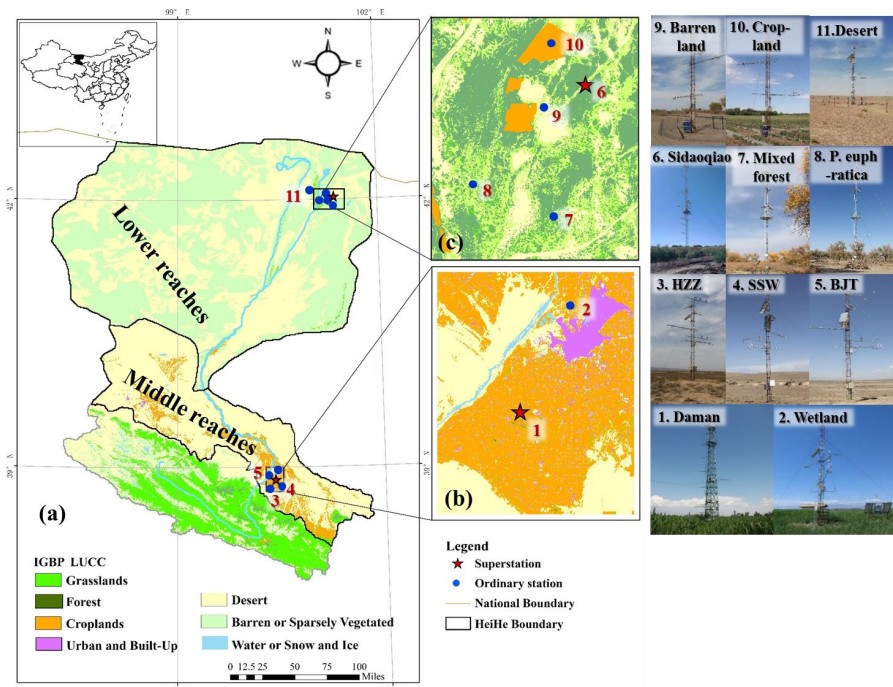


Fig. 1. The middle and lower reach observation systems in the HRB. (a: Heihe River
basin; b: Stations in the Zhangye artificial oasis-desert area in the middle reaches; c:



Stations in the Ejina natural oasis area in the lower reaches)

Table 1. Station information in the middle and lower reaches of the HRB

| ID | Name | Longitude (°, E) | Latitude (°, N) | Elevation (m) | Land Cover | Duration | Location |
|---|---|---|---|---|---|---|---|
| 1 | Daman | 100.37 | 38.86 | 1556 | Maize | May 2012-present | Oasis in midstream, superstation |
| 2 | Zhangye Wetland | 100.45 | 38.98 | 1460 | Wetland mainly reed | June 2012-present | Oasis in midstream, ordinary station |
| 3 | Huazhaizi Desert Steppe | 100.32 | 38.77 | 1731 | *Kalidium foliatum* | June 2012-present | Desert in midstream, ordinary station |
| 4 | Shenshawo Sandy Desert | 100.49 | 38.79 | 1594 | Sandy | June 2012-Apr.2015 | Desert in midstream, ordinary station |
| 5 | Bajitan Gobi | 100.30 | 38.92 | 1562 | Reaumuria | May 2012-Apr.2015 | Desert in midstream, ordinary station |
| 6 | Sidaoqiao | 101.14 | 42.00 | 873 | *Tamarix* | July 2013-present | Oasis in downstream, superstation |
| 7 | Mixed Forest | 101.13 | 41.99 | 874 | *Populus euphratica and Tamarix* | July 2013-present | Oasis in downstream, ordinary station |
| 8 | Populus euphratica | 101.12 | 41.99 | 876 | *Populus euphratica* | July 2013-Apr.2016 | Oasis in downstream, ordinary station |
| 9 | Barren Land | 101.13 | 42.00 | 875 | Bare land | July 2013-Mar.2016 | Oasis in downstream, ordinary station |
| 10 | Cropland | 101.13 | 42.00 | 875 | Melon | July 2013-Nov.2015 | Oasis in downstream, ordinary station |
| 11 | Desert | 100.99 | 42.11 | 1054 | Reaumuria | Apr.2015-present | Desert in downstream, ordinary station |

**2.2 Observation systems**
**2.2.1 Artificial oasis and desert areas in the middle reaches**

The middle reaches are located in the Zhangye oasis in Zhangye City of Gansu

Province, and the primary underlying surfaces include cropland (maize), shelterbelt,
orchard, residential area and wetland (reed) in the oasis and sandy desert, desert steppe



(*Kalidium foliatum*), and the Gobi Desert (Reaumuria) in the surrounding desert. Five
stations (one superstation and four ordinary stations) were established in these surfaces,
which are representative of the main underlying surface types within the oasis-desert
area in the middle reaches of the HRB.
There is one superstation (maize and shelterbelt) and one ordinary station (wetland
and reed) in the Zhangye oasis surrounding three ordinary stations in the desert located
in the middle reaches of the HRB (Fig. 2a). The superstation includes a multiscale
observation system for energy, water vapor and carbon fluxes (lysimeter-EC system-
scintillometer for meter-hundred-kilometer observation scale) and soil moisture
measurements (*in situ* soil moisture profile-cosmic ray probe-soil moisture wireless
sensor network for meter-hundred-kilometer observation scale), and it includes a
hydrometeorological gradient observation system to monitor the profile (7 layers) of
wind speed/direction, air temperature/humidity and carbon dioxide and water vapor
concentration, one layer four-component radiation, air pressure, precipitation, and
infrared temperature (2 repetitions), 9/8 layers' soil temperature/moisture profile, soil
heat flux (3 plates with two buried under the bare soil between two corn plants and one
buried under the corn plants), etc. The EC and hydrometeorological gradient
observation system were installed on a 40 m tower. Optical and microwave
scintillometers were installed on both sides of the 40 m tower apart from 1854 m. There
were also observations of vegetation parameters in the 40 m tower, including a visible
and near infrared phenological camera to monitor the vegetation index and crop growth
curve, two photosynthetically active radiation (PAR) sensors to monitor PAR, a



vegetation chlorophyll fluorescence observation system to monitor sun-induced
chlorophyll fluorescence (SIF), and an LAI wireless sensor network (28 nodes) to
monitor multipoint LAI in the source area of the scintillometer (Fig. 2b, Fig. 4a).

The ordinary stations are comprised of an EC system, an automatic weather station

(AWS) and a visible and near infrared phenological camera. The observation elements
of the AWS include two layers' air temperature/humidity and wind speed/direction, one
layer's four-component radiation, air pressure, and infrared temperature (2 repetitions),
two layers' precipitation, 8/7 layers' soil temperature/moisture, soil heat flux (3 plates),
etc. (Fig. 2c).

The sonic anemometers of the ECs were installed at a height of approximately 3-7 m

above the canopy to capture the sensible heat, latent heat, carbon dioxide and methane
(in wetland) fluxes, etc. The sonic anemometers of all the ECs were aimed toward the
north. Soil parameters, such as soil texture, porosity, bulk density, saturated hydraulic
conductivity, and soil organic matter content, etc. were investigated at each station in
2012 and 2020. Detailed information can be found in Table 2.



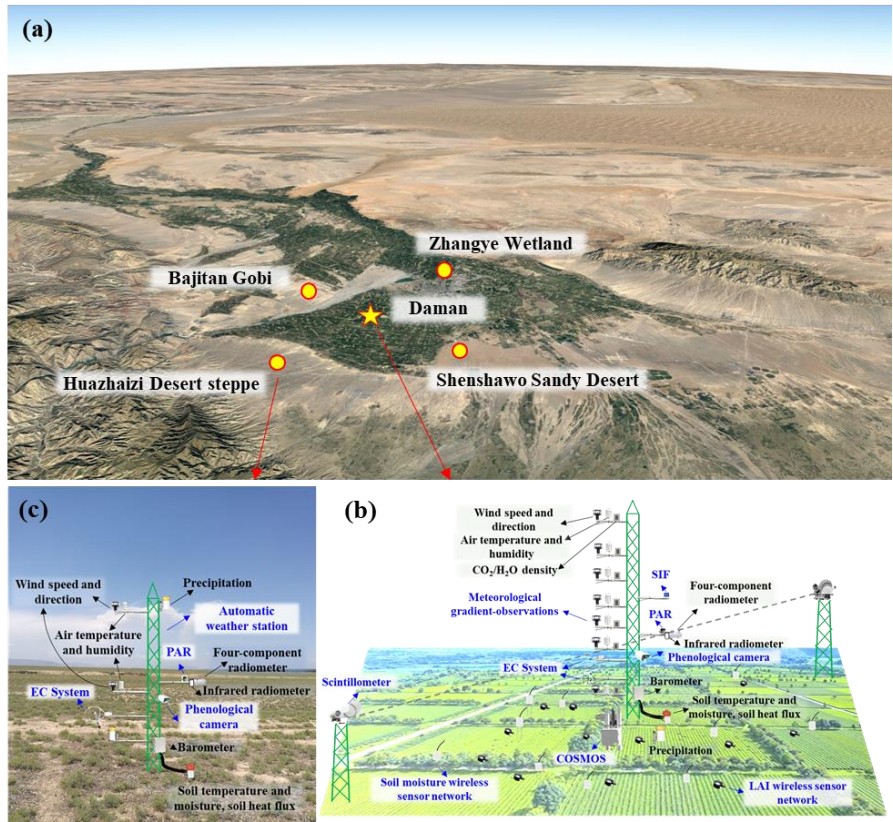


Fig. 2. Sketch map of the artificial oasis and desert area in the middle reaches (a:
artificial oasis and desert area (from © Google Earth); b: Daman superstation; c:
Huazhaizi ordinary station)

### 2.2.2 Natural oasis and desert areas in the lower reaches

The Ejin Banner oasis is located in the lower reaches of the HRB and belongs to
Inner Mongolia and part of Jiuquan city of Gansu Province, which is surrounded by
widespread desert. The main underlying surfaces were Reaumuria and terminal lake in
desert, riparian forest, cropland, barren land and residential area in the oasis in the lower
reaches. There were six stations (one superstation and five ordinary stations) in the



lower reaches, which are located in these land covers, including *Populus euphratica*,
*Tamarix chinensis*, cropland, barren land, and desert.

In the oasis-desert area of the lower reaches, there is one superstation and four

ordinary stations in the oasis and one ordinary station in the desert (Fig. 3a). The
superstations include a multiscale observation system for energy, water vapor and
carbon fluxes (sap flow gauge-EC-large aperture scintillometer for meter-hundred-
kilometer observation scale) and soil moisture measurements (*in situ* soil moisture
profile and cosmic ray probe for meter and hundred meter observation scale), a
hydrometeorological gradient observation system to monitor the profile (6 layers) of
wind speed/direction, air temperature/humidity, one layer four-component radiation, air
pressure, and infrared temperature (2 repetitions), two layers of precipitation, 10/9
layers soil temperature/moisture profile, soil heat flux (with two buried under the bare
soil and one buried under the *Tamarix* plants), etc. The EC and hydrometeorological
gradient observation system were installed on a 28 m tower. Two groups of large
aperture scintillometers were installed on both sides of the 28 m tower apart from 2350
m. The vegetation parameter observations included PAR and the phenological camera
to monitor the vegetation index and crop growth curve installed in the 28 m tower and
LAI wireless sensor network (11 nodes in the source area of the scintillometer) (Fig. 3b,
Fig. 4b). The ordinary stations are comprised of an EC system, an AWS and a visible
and near infrared phenological camera. (Fig. 3c).

Additionally, thermal infrared radiometers and imagers were installed at the Mixed

Forest and Sidaoqiao stations to measure different component temperatures, i.e., the



brightness temperature of different land cover types under different illumination
conditions (Li et al., 2019). The soil parameters and groundwater table were observed
around the stations. Detailed information can be found in Table 2.

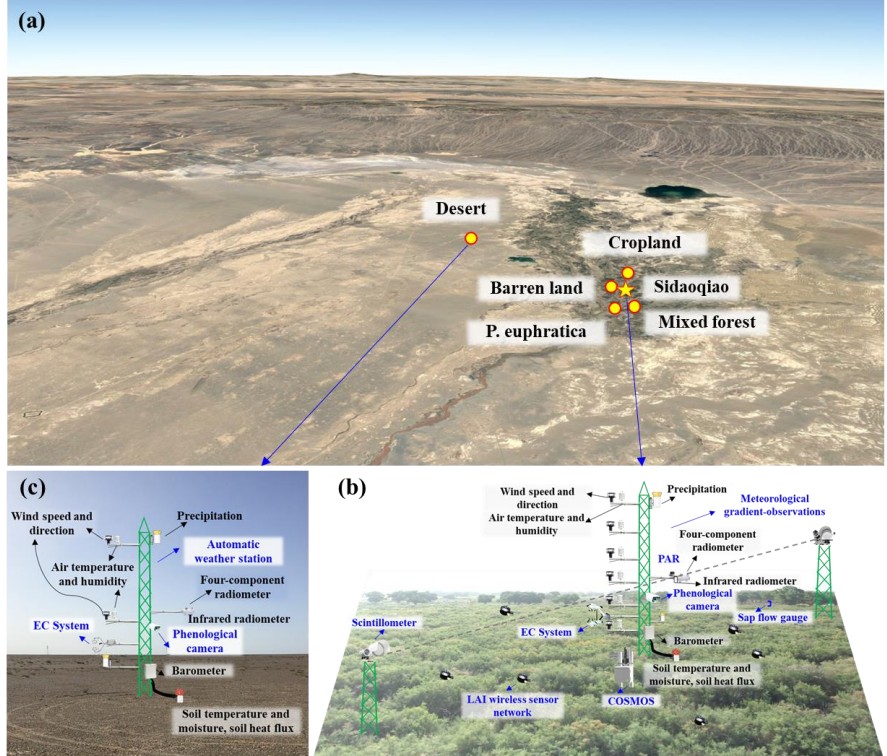

Fig. 3 Sketch map of the natural oasis and desert areas in the lower reaches (a: natural
oasis and desert area (from © Google Earth); b: Sidaoqiao superstation; c: desert ordinary
station)
Table 2 Observation variables and sensor configurations of surface flux,
hydrometeorology, vegetation and soil parameters

| Observations | Sensor | Manufactory | Height/depth (m) | Sites |
|---|---|---|---|---|
| *Surface flux observations:* | | | | |
| Sensible heat, latent heat, carbon dioxide, methane flux | CSAT3&Gill, Li7500&Li7500 A&Li7500DS& EC155&Li7700 | Campbell and LiCor, USA | 3~7 m above the canopy | All stations (methane observation only in wetland, closed path EC at Daman and Desert, |



| | CPEC310 | | | stations) |
|---|---|---|---|---|
| Sensible and latent heat flux | BLS900 and MWSC-160 | Scintec and RPG, Germany | 23.92 | Daman |
| | BLS900 | Scintec, Germany | 25.5 | Sidaoqiao |
| Sap flow | TDP 30 | Rainroot, China | 1.5 | Mixed forest |

*Hydrometeorological observations:*

| | | | | |
|---|---|---|---|---|
| Pressure | PTB110 | Vaisala, Finland | -- | Bajitan, Shenshawo |
| | AV-410BP | Avalon, USA | -- | Mixed Forest |
| | PTB210 | Vaisala, Finland | -- | Huazhaizi |
| | CS100 | Campbell, USA | -- | Daman, Wetland, Sidaoqiao, Desert |
| Precipitation | TE525MM | Texas Electronics, USA | -- | Daman, Wetland, Huazhaizi, Bajitan, Shenshawo,Sidaoqiao, Desert |
| | 52203 | RM Young, USA | -- | Mixed Forest |
| Wind speed/direction | Windsonic | Gill, UK | 3,5,10,15,20,30,40 | Daman, Sidaoqiao |
| | | | 5,7,10,15,20,28 | |
| | | | 5,10 | Wetland, Huazhaizi, Mixed forest |
| | 010C/020C | Met One | 5,10 | Wetland, Desert |
| | 03001 | RM Young, USA | 10 | Bajitan,Shenshawo |
| | | | 28 | Populus euphratica |
| Air temperature/humidity | HMP45D | Vaisala, Finland | 28 | Mixed Forest |
| | HC2S3 | Vaisala, Finland | 5,7,10,15,20,28 | Sidaoqiao |
| | HMP45AC | Vaisala, Finland | 5,10 | Bajitan,Huazhaizi, Shenshawo,Wetland, Desert |
| | | | 28 | Populus euphratica |
| | AV-14TH | Avalon | 3,5,10,15,20,30,40 | Daman |
| Four-component | CNR4 | Kipp&Zonen, Netherland | 10 | Sidaoqiao |
| | | | 22 | Mixed Forest |

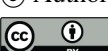



| | | | | |
|---|---|---|---|---|
| radiation | | | | |
| | CNR1 | Kipp&Zonen, Netherland | 6 | Wetland,Huazhaizi,Bajitan,Shenshawo,Desert,Barren land, Cropland |
| | | | 22 | Populus euphratica |
| | PSP&PIR | Eppley, USA | 12 | Daman |
| Infrared temperature | SI-111 | Apogee, USA | 12 | Daman |
| | | | 10 | Sidaoqiao |
| | | | 22 | Populus euphratica, Mixed Forest |
| | | | 6 | Wetland,Huazhaizi,Bajitan,Shenshawo,Desert,Barren land, Cropland |
| Soil temperature profile | 109ss-L | Campbell, USA | 0,-0.02,-0.04,-0.1,-0.2,-0.4,-0.8,-1.2,-1.6,-2.0 | Sidaoqiao, |
| | | | 0, -0.02,-0.04,-0.1,-0.2,-0.4,-0.6,-1.0 | Desert Wetland |
| | AV-10T | Avalon, USA | 0, -0.02,-0.04,-0.1,-0.2,-0.4,-0.6,-0.8,-1.2,-1.6 | Daman |
| | | | 0, -0.02,-0.04,-0.1,-0.2,-0.4,-0.6,-1.0 | Bajitan, Huazhaizi |
| | | | 0, -0.02,-0.04,-0.1,-0.2,-0.4 | Wetland |
| | | | 0,-0.02,-0.04,-0.1,-0.2,-0.4,-0.6,-1.0,-1.6,-2.0,-2.4 | Mixed forest |
| | 109 | Campbell, USA | 0, -0.02,-0.04,-0.1,-0.2,-0.4,-0.6,-1.0 | Shenshawo |
| | | | 0, -0.02,-0.04 | Barren land, Cropland, Populus euphratica |
| Soil moisture profile | ECH$_2$O-5 | Decagon Devices, USA | -0.02,-0.04,-0.1,-0.2,-0.4,-0.6,-1.0 | Bajitan |
| | CS616 | Campbell, USA | | Shenshawo, Desert |
| | | | -0.02,-0.04,-0.1,-0.2,-0.4,-0.6,-0.8,-1.2, -1.6 | Daman |
| | ML2X | Delta-T, UK | -0.02,-0.04,-0.1,-0.2,-0.4,-0.8,-1.2,-1.6,-2.0 | Sidaoqiao |
| | | | -0.02,-0.04,-0.1,-0.2,-0.4,-0.6,-1.0,-1.6,-2.0,- | Mixed Forest |





| | | | 2.4 | |
|---|---|---|---|---|
| | | | -0.02, -0.04 | Barren land, Populus euphratica, Cropland |
| | ML3 | Delta-T, UK | -0.02,-0.04,-0.1,-0.2,-0.4,-0.6,-1.0 | Desert, Huazhaizi |
| Soil heat flux | HFP01 | Hukseflux, Netherland | -0.06 | Wetland,Huazhaizi, Bajitan,Shenshawo,Desert,Barren land, Cropland |
| | HFT3 | Campbell, USA | | Bajitan, Populus euphratica, Mixed forest |
| | HFP01SC | Hukseflux, Netherland | | Daman, Sidaoqiao |
| Averaged temperature | TCAV | Campbell, USA | -0.02, -0.04 | Daman, Sidaoqiao |
| $CO_2/H_2O$ profile | AP200 | Campbell, USA | 3,5,10,15,20,30,40 | Daman |
| Groundwater Table | U20 | Onset, USA | -2~-3m | Sidaoqiao, Mixed forest, Populus euphratica, Cropland, Desert |
| *Vegetation parameter observations:* | | | | |
| Vegetation phenology | Phenological camera | XST-PhotoNet, China | above the canopy | All sites |
| LAI | XST- LAINet | Beijing StarViewer Science and Technology Ltd., China | Below the canopy | 28 nodes around Daman, 6 nodes around Sidaoqiao, 5 nodes around Mixed forest |
| photosynthetically active radiation | PQS-1 | Kipp&Zonen, Netherland | 0.5, 12 | Daman |
| | | | 10 | Sidaoqiao |
| | | | 22 | Mixed forest, Populus euphratica |
| | | | 6 | Wetland, Cropland |
| Sun-induced chlorophyll fluorescence | AutoSIF-1 | Beijing Bergsun Spectral Technology Co. Ltd, China | 34 | Daman |
| *Soil parameters*: soil sampling and laboratory testing in 2012 and 2020 | | | | |


## 3. Data processing and quality control

The data processing and quality control procedure can be divided into data collection,



data processing and data archiving and sharing (Fig. 4).
In the data collection step, the comparison and calibration of instruments are
prerequisites to ensure the quality of the observation data. The instrument comparison
experiments were specifically arranged under the Gobi Desert in 2012 in the middle
reaches (Xu et al., 2013) and shrub in 2013 in the lower reaches (Li et al., 2018) to
ensure the consistency and comparability of the instruments. In addition, the
instruments with multiple layers were compared at the same height before installation,
the soil moisture probes were also compared under dry and wet conditions, and the
multitype rain gauges were compared in the same field. The infrared gas analyzer of all
the EC systems was calibrated at the beginning and end of the vegetation-growing
season every year. To ensure the data quality, a routine maintenance procedure is
formulated and strictly followed, including daily (checking the real-time data through
remote monitoring and data management system for field observatory network v1.0),
10 days (checking the time series plot providing by the system), monthly (routine
inspection in every station), and annually (data processing and release) (Liu et al., 2018).
The Heihe watershed internet of things observation system was developed to complete
the above maintenance procedure, which included remote receiving and storing the filed
data, browsing and processing real-time data, monitoring the instrument status and early
warning the abnormal conditions, etc.
During the data processing step, a processing scheme was formulated for each type
of instrument. For the EC system, the data were processed from the raw 10 Hz turbulent
data, including spike detection, sonic temperature correction, coordinate rotation,





frequency response correction, and WPL (Webb-Pearman-Leuning) correction. (Liu et
al., 2016; Wu et al., 2023). Additionally, the 30-min flux data series were identified as
quality flags according to the stationarity test and integral turbulence characteristics test.
A final quality flag (1~9) was assigned to each specific turbulent flux value, indicating
good quality (1~3), suitability for general use (4~6), poor but better than gap filling data
(7~8), and discarded data (9). The data processing steps from scintillometer
measurements to surface fluxes are as follows: raw data to light intensity variance, light
intensity variance to the structure parameter of the refractive index of air ($C_n^2$), $C_n^2$ to
meteorological data, and finally obtaining surface fluxes combining the meteorological
data. Four steps are taken to ensure the quality of scintillometer data (Liu et al. 2011;
Zheng et al., 2023): (i) excluding data for $C_n^2$ beyond the saturation criterion; (ii)
excluding data obtained during periods of precipitation; (iii) excluding data when the
demodulated signal is small; and (iv) excluding data when the sensor is malfunctioning.
The steps of the meteorological gradient observation system and AWS data processing
and quality control were twofold: (1) all the AWS data were averaged over an interval
of 10 min for a total of 144 records per day. The missing data were denoted by -6999;
(2) the unphysical data were rejected, and the gaps were denoted by -6999. The surface
soil heat flux was calculated using the 'PlateCal' approach (Liebethal et al., 2005), and
the final surface soil heat flux was the weighted vegetation fraction combined with the
soil temperature and moisture measured above the heat plates. The vegetation growth
curve and vegetation index can be obtained from visible and near infrared bands
measured by phenological cameras. The key phenological parameters are determined



according to growth curve fitting, such as the growth season start date, peak, and growth
season end. The leaf area index (LAI) data were obtained from the LAINet sensor,
which can continuously measure the multipoint total solar radiation above the canopy
and the transmitted radiation below the canopy, and the LAI was calculated based on
multiangle transmittance data (Qu et al., 2014). Then, all the data are processed into a
standardized file for sharing.
During the archiving and sharing step, the metadata were written for each data point,
including the site description, processing step, header description, and other notes. (Li
et al., 2017a). Before data are released, self-examination, crosschecks and expert review
are required to ensure data quality. Finally, the data were archived and shared online.

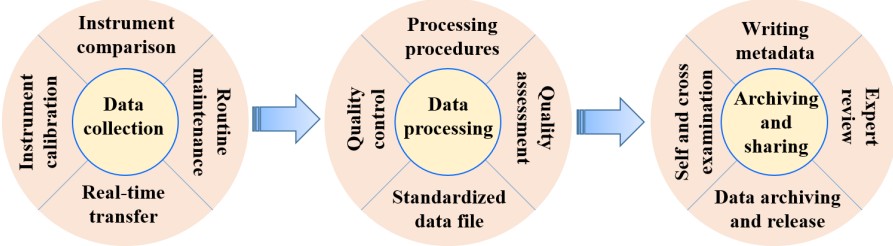

Fig. 4 Flowchart of data processing and quality control

**4.  Data description**
**4.1 Energy, water vapor and carbon fluxes data**
The EC systems were used to measure surface flux at all sites, namely, 5 stations (2
in oasis, 3 in desert) in the middle reaches and 6 stations (5 in oasis, 1 in desert) in the
lower reaches. The turbulent flux data were recorded by the open path or closed path
EC systems and processed carefully. In addition to the surface flux of sensible, latent
and carbon dioxide, the methane flux was also observed at the wetland site in the middle
reaches (Table 2). The multiyear seasonal variations in sensible heat, latent heat, carbon





dioxide and methane fluxes are shown in Figure 5. Generally, the latent heat fluxes in
oases are obviously higher than those in deserts, especially in the lower reaches. The
latent heat fluxes exhibited a single peak during one year, with a peak value of
approximately 200 W m$^{-2}$ in the oasis area; however, they significantly fluctuated due
to irrigation (normally 4 times in cropland of the midstream region, 2 times in riparian
forest and melon of the downstream region) or precipitation. In the middle reaches, the
latent heat flux in the wetland showed the largest values, which were more than 200 W
m$^{-2}$ in the crop growing season, and it also presented relatively large values in the
midstream piedmont desert region with dense *Kalidium foliatum* cover (peak value
greater than 50 W m$^{-2}$). In the lower reaches, the latent heat flux showed consistent
variations in the riparian forest with a peak value of approximately 150 W m$^{-2}$ during
the crop growing season; however, it showed large fluctuations in the melon surface
during growth due to frequent irrigation (approximately 7~8 times), and the bare land
in the oasis and desert had a small latent heat flux.
The seasonal variations in sensible heat flux were totally different in the oasis and
desert systems. The sensible heat flux showed two peaks in the oasis in both the middle
and lower reaches except for the bare land, namely, reaching maximum values at the
end of April and September, and it showed minimum values in mid-August (-25 W m$^{-}$
$^{2}$), corresponding to large values of latent heat flux in the oasis that were even greater
than net radiation. This phenomenon was also found by previous researchers (Liu et al.,
2011) and was called the 'oasis effect'. In the desert area, the sensible heat flux appeared
as a single peak in spring and decreased gradually since then. The variation in sensible
heat flux in bare land of the natural oasis in the lower reaches is similar to that in the
desert area.
In the oasis, the carbon dioxide ($CO_2$) flux showed obvious 'U' variations, especially





in the middle reaches. The crop absorbed carbon dioxide (carbon sink) in the crop-
growing season, and a negative value of approximately -14 µmol m$^{-2}$ s$^{-1}$ was observed
in the maize surfaces. The magnitude of the methane ($CH_4$) flux was lower than the
$CO_2$ flux and was in the range of approximately 0~0.1 µmol m$^{-2}$ s$^{-1}$ in the wetland. The
$CH_4$ flux in the non-growing season was the lowest and increased rapidly in April.
Although the magnitude of the $CH_4$ flux was lower than the $CO_2$ flux, the contribution
of methane emissions to global warming was as important as $CO_2$ contributions on a
long time scale (Hommeltenberg et al., 2014; Zhang et al., 2016b), especially focusing
on $CH_4$ flux measurements in wetlands (Zhang et al., 2022). The variations in $CO_2$ flux
in the riparian forest were relatively small, with values of approximately -0.4 µmol m$^{-2}$
s$^{-1}$ in the plant growing season. There was little carbon sequestration in the desert area
due to little or sparse vegetation, and the $CO_2$ flux in the desert area was very small,
fluctuating around zero during the years.

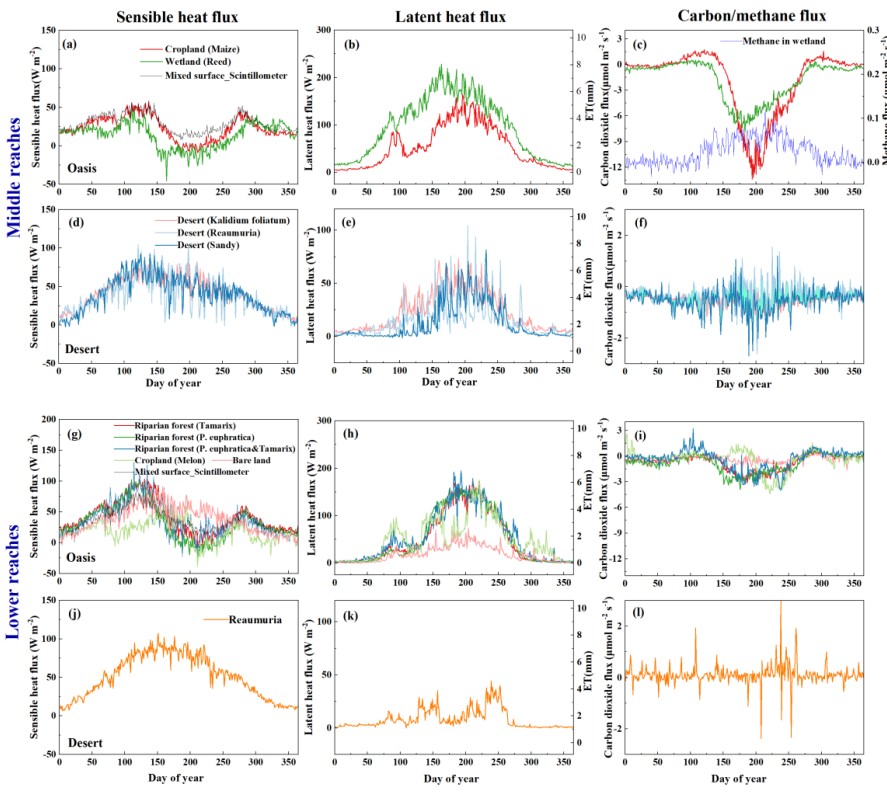

Fig. 5 The multiyear seasonal variations in sensible, latent heat, carbon dioxide and
methane fluxes in the oasis-desert area (sensible heat flux—left, latent sensible heat
flux—middle, carbon dioxide and methane flux—right, 2012-2021)

## 4.2 Hydrometeorological data

The hydrometeorological data were obtained from 13 AWSs, with six in the middle
reaches (Fig. 2) and seven in the lower reaches (Fig. 3) of the HRB. All the AWSs
recorded four-component radiations (short/long wave upward and downward radiation),
soil heat flux, surface and soil temperature profiles, air temperature and humidity, wind
speed and direction, air pressure, precipitation, soil moisture profiles, and groundwater



table, etc. (Table 2). All sensors were calibrated and intercompared before being
mounted. The sampling frequencies, reference heights and directions of these sensors
at all stations were identical to maintain consistency.
4.2.1 Radiation, soil heat flux, surface and soil temperature profile
It is important to understand the variations in radiation and surface soil heat flux in
oasis and desert areas, which are the surface available energy. Figure 6 shows the four
radiation components and soil heat flux in oasis and desert areas in the middle and lower
reaches in the HRB, and all the variables exhibited obvious seasonal variations with an
inverted 'U' shape. The incoming shortwave radiation was consistent with each other
in oasis and desert because of the short distance among the sites. Due to the higher
albedo in the desert, the upward shortwave radiation in the desert was larger than that
in the oasis (approximately larger than 30%). The incoming longwave radiation
originates from the atmosphere (in particular $CO_2$ and water vapor) and thermal
radiation of clouds in the lower atmosphere. The oasis presents relatively large water
vapor and cloudiness; thus, the incoming longwave radiation for the oasis was greater
than that for the desert (approximately 2%). It is to be expected that under dry
conditions during the daytime, the surface temperature of the desert will be significantly
greater than that of the well-watered oasis site. Consequently, the upward longwave
radiation in the desert was larger than that in the oasis (approximately 8%). The net
radiation, driving the turbulent fluxes of sensible heat and latent heat at the earth surface
and heating soil, was greater in the artificial oasis and the natural oasis than in the desert
at approximately 50 W m$^{-2}$. The daily mean surface soil heat fluxes varied similarly in
oasis and desert areas with relatively low values in the range of -20 to 20 W m$^{-2}$.

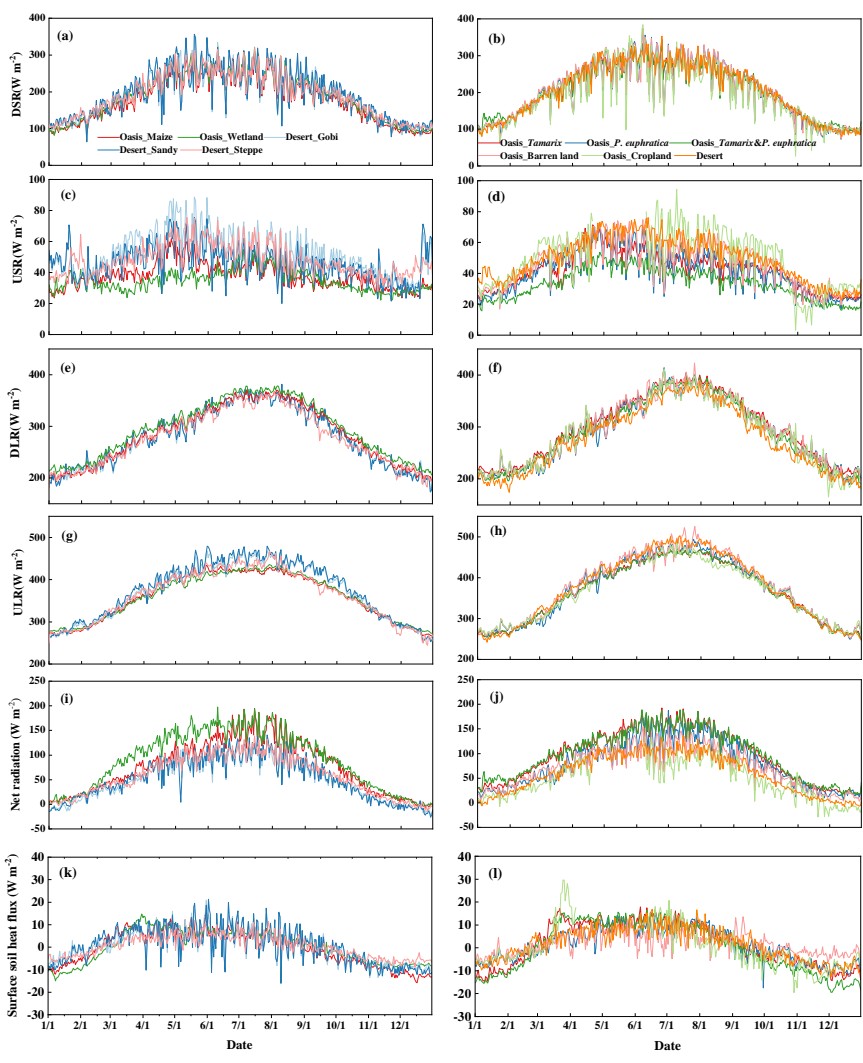


Fig. 6 Seasonal variations in multiyear average radiation components in the oasis-desert
system (middle reaches: a, c, e, g, i, k; lower reaches: b, d, f, h, j, l; 2012-2021 daily
averaged DSR: downward shortwave radiation; USR: upward shortwave radiation;
DLR: downward longwave radiation; ULR: upward longwave radiation)
The soil temperature exhibited a signal peak around the year in the range of -
15°C~34°C, and it decreased with increasing soil depth during the plant growing season;
however, it exhibited an increasing trend in the winter. The shallow soil began to thaw
at the beginning of spring (march) and to freeze in autumn (November). The soil
temperature changed little with depth when it exceeded 0.8 m and 1 m in the oasis and
desert, respectively. The soil temperature in the desert was significantly higher by
approximately 10 ℃ during the plant growing season than that in the oasis in both the
middle and lower reaches. Additionally, the soil temperature in the artificial oasis-desert
area (middle reaches) was approximately 5 ℃ lower during the plant growing season
than that in the natural oasis-desert area (lower reaches) (Fig. 7).

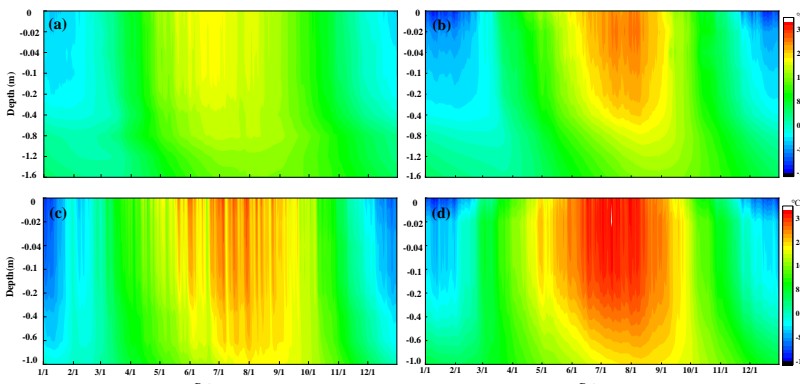


Fig. 7. Seasonal variations in surface and soil temperature profiles in oasis and desert
areas (2012-2021) (a: oasis in middle reaches–maize; b: oasis in lower reaches–*Tamarix*;
c: desert in middle reaches–Reaumuria; d: desert in lower reaches–Reaumuria)
4.2.2 Air temperature/humidity, wind speed/direction, air pressure
To show micrometeorological characteristics clearly, the comparison of daily average
air temperature and relative humidity (5 m except the *P. euphratica* surface with a height
of 28 m), wind speed (10 m) and air pressure in desert and oasis are plotted in Figure 8.
The seasonal variation in air temperature in the oasis and desert was similar; however,





the air temperature in the desert was generally higher than that in the oasis by
approximately 0.6 °C on average annually (approximately 0.4 °C in the plant growing
season). Instead, the relative humidity in the desert was lower than that in the artificial
oasis in the midstream region (approximately 9% and 10% in the annual and plant
growing seasons, respectively). The relative humidity in natural oasis and desert areas
are similar due to the extreme arid regions with rare precipitation, little irrigation
amount and small natural oasis area. Generally, the desert surface has the characteristics
of high temperature and lower humidity, and the oasis is a cold and wet island. In the
middle and lower reaches of the oasis and desert areas, the wind speed in the desert was
obviously larger than that in the oasis because of the wind shield effect in the oasis
(middle reaches: 1~3 m/s in the oasis, 2~6 m/s in the desert; lower reaches: 3~6 m/s in
the oasis, 3~7 m/s in the desert), and the wind speed decreased significantly when
passing by the windbreaks, buildings and crops, especially in the artificial oasis in the
middle reaches. The lower wind speed in oases is helpful to plant growth, people's
survival environment and the maintenance of oasis and desert ecosystems (Wang and
Cheng, 1999). While the seasonal variation in wind speed between desert and oasis was
similar, this indicated that they were controlled by the same synoptic system. The wind
speed in the natural oasis in the lower reaches was higher than that in the artificial oasis
in the middle reaches. The maximum wind speeds were observed in April in the
artificial and natural oases, respectively, while the minimum values were observed in
July. The air pressure decreased with decreasing elevation, e.g., the air pressure in the
middle reaches with relative high elevation was lower than that in the lower reaches, as



well as the discrete distribution of stations in the middle reaches with different
elevations (Fig. 10g and h, Table 1).

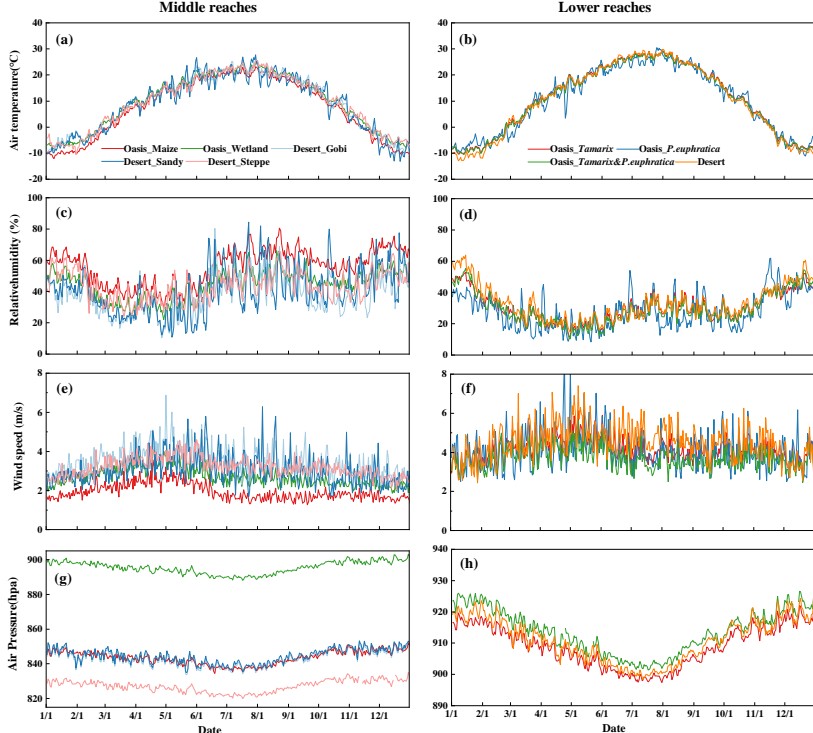

Fig. 8 Micrometeorological comparison between the oasis and desert (a, b: air
temperature; c, d: relative humidity; e, f: wind speed; g, h: air pressure, 2012-2021)
Windbreaks, buildings, crops or riparian forests drag on the wind flow inside the oasis,
and the wind direction is different in the oasis and desert. In the middle reaches, the
dominant wind directions in the desert are the northwest wind and southeast wind
directions, while they are northwest and southwest (10 m) in the oasis cropland;
however, with the increase in observation height, the influence of surface roughness on
wind speed/direction decreased, and the southwest wind gradually decreased, while the
northwest wind and southeast wind gradually increased, which is similar to the wind in



the desert area around the oasis (~30 m height). In the lower reaches, the wind direction
was similar in the oasis and desert areas, with prevailing wind directions of west and
east (Fig. 9).

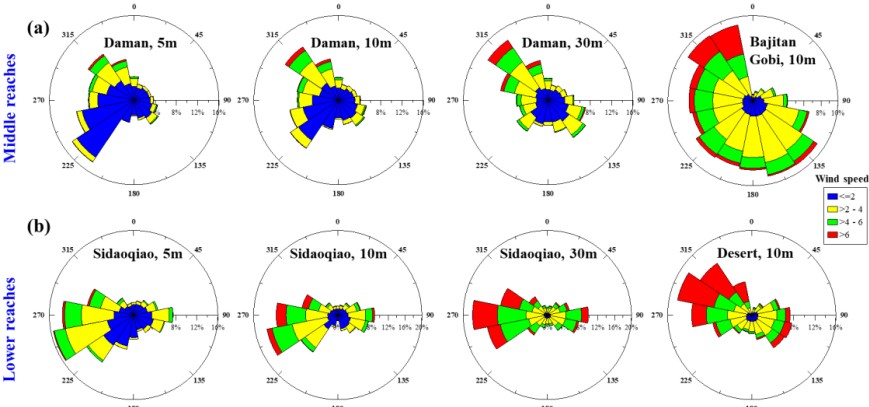


Fig. 9 Wind speed/direction in the oasis and desert area (2012-2021) (a: artificial oasis-
desert area in middle reaches; b: natural oasis-desert area in lower reaches; legend is
wind speed)

There are six/seven layer gradient observations of wind, air temperature and

humidity in superstations in artificial and natural oases. Data on typical days during
January, April, July and October in 2021 were selected, and the profiles of wind speed,
air temperature and humidity are plotted in Fig. 10. The wind speed generally increased
with the observation height, especially in the natural oasis. The air temperature showed
inversion at night during atmospheric stable stratification and changed little even below
10 m in the afternoon in July at both artificial and natural oases, which may be caused
by oasis-desert interactions. The relative humidity was low during the daytime and
maintained high values at night, decreasing with the observation height, especially
below 10 m.

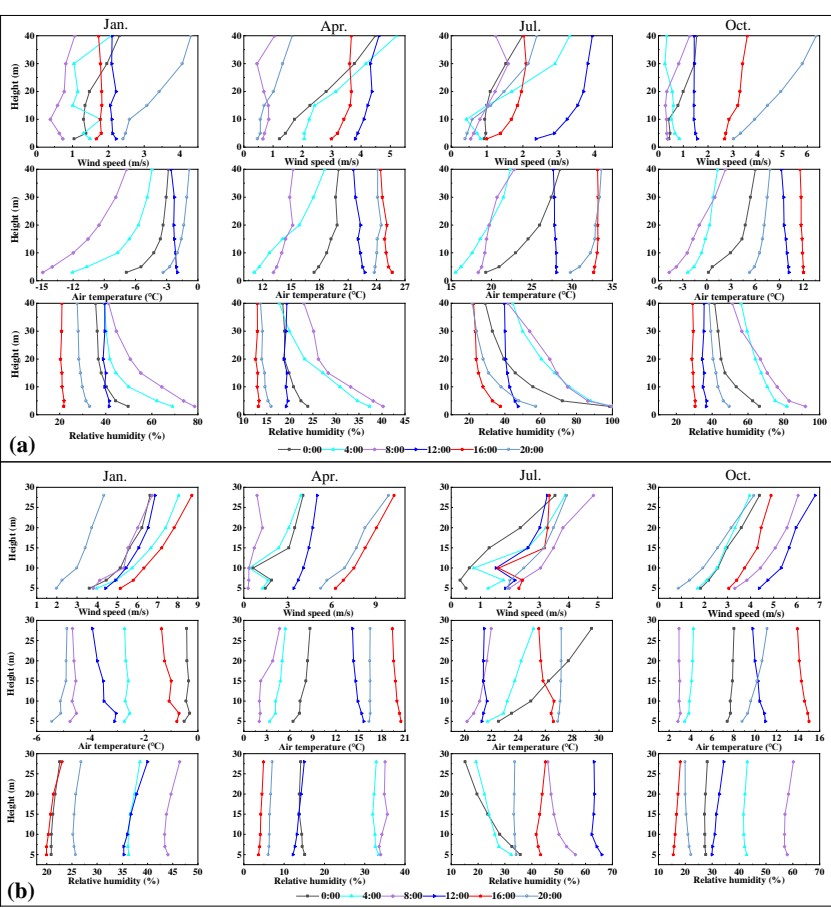

Fig. 10 The profile of wind speed, air temperature and relative humidity in typical days
of January 14, April 14, July 14 and October 14 in 2021 (a: artificial oasis in middle
reaches; b: natural oasis in lower reaches)
4.2.3 Precipitation, soil moisture and groundwater table
Figure 11 shows the variations in precipitation, soil moisture profiles and
groundwater table (lower reaches) in typical oasis and desert ecosystems. Precipitation
in the middle reaches was higher than that in the lower reaches, and it was higher in the



oasis than in the desert. The soil moisture in the oasis was significantly higher than that
in the desert, and it was especially small in the desert of the lower reaches. The soil
moisture exhibited an increasing trend with increasing soil depth, especially in the oasis.
The soil moisture was higher at depths of 0.8-1 m in the artificial oasis in the middle
reaches and at depths of 0.4-0.8 m in the natural oasis in the lower reaches. Soil crust
appeared in the lower reaches due to soil salinization, and it may prevent the loss of soil
moisture. When a precipitation event occurred, the soil moisture in the desert increased
accordingly; however, there were no clear variations in the oasis. There were usually
four irrigation events in the artificial oasis in the middle reaches, and the soil moisture
increased clearly accordingly, while some occasional peaks in soil moisture were due
to relative heavy precipitation (Fig. 11a). In the lower reaches, two irrigation events
(usually in March and September) generally occurred in riparian forests in natural oases.
The shallow soil moisture showed large values in March when irrigation occurred and
decreased in the plant growing season with a slight increase in September. Another
phenomenon is that the precipitation in the artificial oasis was larger than that in the
desert, although the sites were not far away from each other (e.g., 103.1 mm at the
Daman superstation and 75.4 mm at the Gobi station). From the analysis, the soil
moisture in the desert was strongly dependent on precipitation (Fig. 11c, d), while it
maintained high values in the plant growth season relying on irrigation in the oasis.
In the lower reaches, five systems for groundwater table measurement have operated
since June 2014 in the oasis, near the Sidaoqiao, Mixed Forest, *Populus euphratica*,
Cropland, and Barren Land stations. The groundwater table was approximately 1–3 m



under the ground, and the groundwater table level declined from a depth of
approximately 1 m to 3 m in the growing season to supply the riparian forest growth
(Fig. 11b). Additionally, one groundwater table measurement system was installed near
the desert station in 2018. The depth of the groundwater table level was approximately
10-11 m in the desert and showed no significant variation over the years (Fig. 11d).

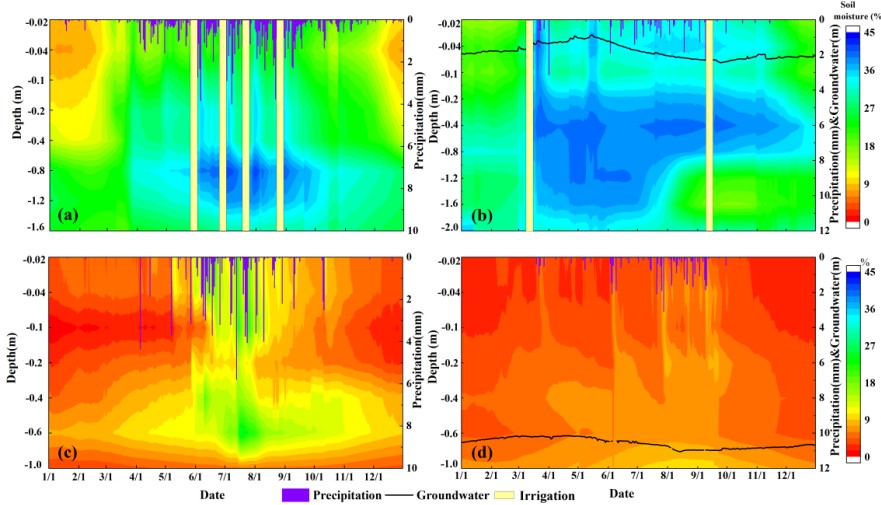


Fig. 11. Comparison of precipitation and soil moisture profile between desert and oasis
(2012-2021, a: oasis in the middle reaches (maize, Daman); b: oasis in the lower reaches
(*Tamarix*, Sidaoqiao); c: desert in middle reaches (Bajitan Gobi); d: desert in lower
reaches (desert))
**4.3 Vegetation and soil parameters**
The vegetation parameters include photosynthetically active radiation (PAR), leaf
area index (LAI), phenology, sun-induced chlorophyll fluorescence (SIF), etc. The PAR,
the amount of light available for photosynthesis, is observed at stations with vegetation



cover, and it can be used as the source of energy for photosynthesis by green plants.
The PAR observations showed similar seasonal variations in typical oasis ecosystems
in the middle and lower reaches, with a maximum daily PAR of approximately 750
$\mu$mol m$^{-2}$ s$^{-1}$ (Fig. 14a). Vegetation parameters, such as LAI and phenology, were also
observed in the middle and lower reaches. LAI in the middle reaches (maize) increased
gradually with crop growth, and it was larger than that in the lower reaches (*Tamarix*),
which showed little change in this shrub surface (Fig. 14b). The phenological camera
was installed at each station except the desert to acquire the phenology. The greenness
index of the green chromatic coordinate (GCC) was derived to capture the key
phenological phase of the plant, such as the SOS (start of season), POP (position of
peak value), and EOS (end of season) (Fig. 14c).

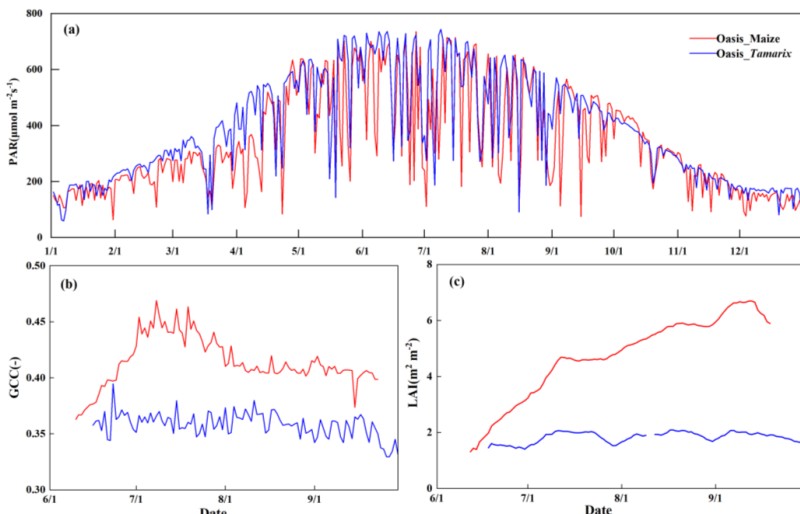


Fig. 12. Variations in vegetation parameters in the middle and lower reaches of the oasis
(a, b, c are PAR, LAI and GCC in the artificial and natural oases, respectively, in 2018)
Soil samples were collected at each station in the middle and lower reaches in 2012



and 2020. These soil samples were analyzed in the laboratory, and parameters such as
soil texture, porosity, bulk density, saturated hydraulic conductivity, and soil organic
matter content were obtained. Some soil parameters at typical stations are shown in
Table 3. Silty soil is dominant in the oasis, and sand is dominant in the desert. The
porosity and bulk density showed no significant difference. The saturated hydraulic
conductivity and soil organic matter at the typical stations are also given in Table 3.
Table 3 Soil parameter measurements at typical stations in 2020

| | Station | Soil texture | Soil properties |
|---|---|---|---|
| Middle reaches | Daman (Oasis) | Clay: 6% Silt: 69% Sand: 25% | Porosity: 47.1 %; Bulk density: 1.46 g/cm$^3$; Saturated hydraulic conductivity: 0.177 mm/min; Saturated water capacity: 64.10 %; PH: 8.48; NH$_4^+$-N: 0.83 mg/kg; NO$_3^-$-N: 15.90 mg/kg; Soil carbon content: 1.85 %; Soil organic carbon content: 0.72 %; Soil nitrogen content: 0.027% |
| | Huaizhaizi (Desert) | Clay: 1% Silt: 19% Sand: 80% | Porosity: 38.0 %; Bulk density: 1.49 g/cm$^3$; Saturated hydraulic conductivity: 4.93 mm/min; Saturated water capacity: 22.21 %; PH: 8.27; NH4$^+$-N: 0.77 mg/kg; NO$_3^-$-N: 29.70 mg/kg; Soil carbon content: 1.83 %; Soil organic carbon content: 0.33 %; Soil nitrogen content: 0.026% |
| Lower reaches | Sidaoqiao (Oasis) | Clay: 21% Silt: 69% Sand: 10% | Porosity: 45.8 %; Bulk density: 1.47 g/cm$^3$; PH: 8.80; NH$_4^+$-N: 1.02 mg/kg; NO$_3^-$-N: 5.23 mg/kg; Soil carbon content: 2.02 %; Soil organic carbon content: 0.70 %; Soil nitrogen content: 0.070% |
| | Desert around terminal lake (Desert) | Clay: 9% Silt: 7% Sand: 84% | Porosity: 44.4 %; Bulk density: 1.49 g/cm$^3$; PH: 8.62; NH$_4^+$-N: 0.26 mg/kg; NO$_3^-$-N: 5.74 mg/kg; Soil carbon content: 1.42 %; Soil organic carbon content: 0.38 %; Soil nitrogen content: 0.039% |

**5. Data availability**
The dataset of energy, water vapor and carbon exchange observations in oasis-desert
areas reported in this study, including energy, water vapor and carbon fluxes,
hydrometeorological data, and vegetation and soil parameters, are available and can be
downloaded freely at the National Tibetan Plateau Data Center
(https://doi.org/10.11888/Terre.tpdc.300441, Liu et al., 2023). A specific directory for
each observation station was designated with data classified into three categories,



namely, energy, water vapor and carbon fluxes, hydrometeorological data, and
vegetation and soil parameter data. Short descriptions were also provided for each
dataset. The Beijing standard time was used in all the data files (UTC+8).
**6. Conclusions**
The typical land covers in the middle and lower reaches over the HRB are oases and
deserts characterized by fragile environments. Oasiszation and desertification are two
opposing processes in arid and semiarid regions with scarce water resources. To combat
desertification around oases and maintain the sustainable development of oases, a land
surface process integrated observatory network was established in the oasis-desert area
in the middle and lower reaches of the HRB. Eleven stations (7 in oasis, 4 in desert)
have been established in these regions since 2012 to monitor the energy, water vapor
and carbon exchange between land and atmosphere over oasis and desert areas, and a
long-term and high-quality oasis and desert dataset of energy, water vapor and carbon
fluxes and auxiliary parameters was produced. This study shows the achievements of
11 stations over 10 continuous years of observations, including energy, water vapor and
carbon fluxes, hydrometeorology, vegetation and soil parameter data. These data can
be used in the analysis of the water-heat-carbon process and its influence mechanism
(Wang et al., 2019; Xu et al., 2020; Bai et al., 2021; Wu et al., 2023), calibration and
validation of remote sensing products (Ma et al., 2018; Song et al., 2018; Li et al., 2021;
Zhang et al., 2022), and simulations of energy, water vapor and carbon exchange (Li et
al., 2017b; Liu et al., 2020; He et al., 2022; Zhou et al., 2022). We confirm that the 10-
year long-term dataset presented in this study is of high quality with few missing data



and believe that the datasets will support ecological security and sustainable
development in oasis-desert areas. Most of the stations are ongoing observations, which
can play a greater role in such ecologically fragile areas and provide a reference for
other similar oasis-desert areas along the Silk Road.

**Acknowledgement**
This work was supported by the Strategic Priority Research Program of the Chinese
Academy of Sciences (Grant no. XDA20100101), the National Natural Science
Foundation of China (42171461).

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
