# Peer review of "A dataset of energy, water vapor and carbon exchange observations in oasis-desert areas from 2012 to 2021 in a typical endorheic basin"

_Earth System Science Data, 2023_

## Author Comment (AC1)

**RC1**: 'Comment on essd-2023-149', Anonymous Referee #1, 31 Jul 2023

"A dataset of energy, water vapor and carbon exchange observations in oasis-desert areas from 2012 to 2021 in a typical endorheic basin" provides a comprehensive description of hydrometeorological data observations over 2012-2021 in the Heihe River Basin in northern China. This includes descriptions of station locations, instrumentation and set up, data collection, and data summaries. The data network is located in a narrow, focused area, but provides a great contrast between desert and oases environments. This is a high-quality data description paper with a strong justification for an observational network in this region. I have very few comments for minor improvements.

Thanks for the constructive comments.

Lines 157-158: could the authors provide more details about the cryosphere, given the importance for water supply. For example, snow covered season and typical snowpack magnitude.

Response: The upper reaches of Heihe river basin is the water supply to the middle and lower reaches. In the upper reaches, glaciers, snow cover and frozen ground is widely distributed, and snowfall could occur in any season in the high-altitude mountainous regions (elevation >3800 m). In January and February, snowfall events were rare in historical records due to the relatively low moisture in the atmosphere, while spring and autumn (i.e., March to May, October to November) are considered two main snowfall seasons. The typical snow depth is 15-30 cm with a duration of 90-120 days in the snow-covered regions. However, drifting snow is also commonly observed in the region, which may lead to the redistribution of snowpack in high-elevation regions (Che et al., 2019). The observations in the upper reaches of Heihe river basin were described in Che et al. (2019) and observation data were also released.

We have also added related descriptions in the revised manuscript in lines 158-161. "In the upper reaches, glaciers, snow cover and frozen ground is widely distributed and snowfall could occur in any season (elevation >3800 m). The typical snow depth is 15-30 cm with a duration of 90-120 days in the snow-covered regions (Che et al., 2012; Che et al., 2019)."

Line 167: recommend revising to (Table 1; Section 2.2) so that the reader knows where 'superstation' is defined

Response: We have revised accordingly. Please see lines 170-171. "…regions since 2012 with two superstations and eleven ordinary stations (Table 1; Section 2.2), …"

Line 217 and 253: Figures 2 and 3. It would be useful to provide a scale on this map so the reader can estimate distance between stations

Response: These two figures (Fig. 2 and 3) have been revised accordingly.

Lines 315; 216: This is a great flowchart!

Response: Thanks!

Line 373: What is included in 'etc.?' and can that be added to the list rather than referred to as 'etc'

Response: We have revised accordingly. Please see lines 385-386. "…speed and direction, air pressure, precipitation, soil moisture profiles, infrared temperature, and groundwater table in the lower reaches."

Lines 376, 415, 475: subtitles 4.2.1, 4.2.2, and 4.2.3 should be bold

Response: Done.

References:

Che, T., Dai, L.Y., Wang, J., Zhao, K., and Liu, Q: Estimation of snow depth and snow water equivalent distribution using airborne microwave radiometry in the Binggou Watershed, the upper reaches of the Heihe River basin, Int. J. Appl. Earth Obs., 17, 23-32, https://doi.org/10.1016/j.jag.2011.10.014, 2012.

Che, T., Li, X., Liu, S.M., Li, H.Y., Xu, Z.W., Tan, J.L., Zhang, Y., Ren, Z.G., Xiao, L., Deng, J., Jin, R., Ma, M.G., Wang, J., and Yang, X.F.: Integrated hydrometeorological, snow and frozen-ground observations in the alpine region of the Heihe River Basin, China, Earth Syst. Sci. Data, 11, 1483–1499, 2019.

---

## Author Comment (AC2)

**RC2**: 'Comment on essd-2023-149', Anonymous Referee #2, 02 Aug 2023

This study with a title of "A dataset of energy, water vapor and carbon exchange observations in oasis-desert areas from 2012 to 2021 in a typical endorheic basin" has been seriously reviewed. Overall, this paper is well organized, including written English and structures. Importantly, I believe that this dataset can provide valuable data to explore the water-heat-carbon process and its influence mechanism, calibrate and validate related remote sensing products, simulate energy, water vapor and carbon exchange in oasis and desert areas. Now, I have only several suggestions for this study, and recommended it to be accepted after a minor revision.

Response: Thanks for the constructive comments.

1. In section 3, it is better to show more detailed information about the post-processing of the data, e.g., missing values, and energy imbalance (which method was used for this data).

Response: Thanks for the valuable suggestions. As suggested by the reviewer, we have added a more detailed interpretation in section 3, as follows:

lines 291-295 "The unclosed energy balance of EC system is a universal problem. There was approximately an average of 17% energy imbalance in our study area (Xu et al., 2017; Zhou et al., 2018), which was reasonable compared with previous results (Stoy et al., 2013). The Bowen-ratio correction method is recommended to close the energy balance (Twine et al., 2000; Xu et al., 2020)"

lines 310-314 "There are approximately 10–20% missing or rejected values of EC or scintillometer data. The look-up table (LUT) method is recommended to fill the gaps when data were missing (Xu et al., 2020). The maximum missing values of AWS data were no more than 10%, and linear interpolation method is recommended to fill the missing values."

lines 321-322 "Seven days moving averaged method is recommend to eliminate noise from the daily LAI observations (Qu et al., 2014)."

2. In section 4, how did the authors process the data points with missing values before drawing these figures?

Response: Missing data is inevitable in long-term observations due to instrument malfunction, maintenance and calibration, bad weather, power loss, etc. There are approximately 10–20% missing or rejected values of eddy covariance or scintillometer

data. Relatively few missing data (less than 10%) were found in meteorological observations, and the green chromatic coordinate (GCC) and leaf area index (LAI) data were relative continuous. In this manuscript, we mainly draw the figures using the released data which were not filled the gaps. Generally, figures in section 4 were plotted using the days with less or no missing data.

3. Maybe it is better to show figure 10 using the shaded maps (like Figure 11) for each time of a day. Why figure 10 was drawn with data of typical days rather the whole observation period (e.g., multi-year mean)?

Response: Thank you for the suggestions. The study area of this paper is in oasis-desert areas, and the interactions between oasis and desert will change the observations of wind speed, air temperature and humidity gradient, especially in the afternoon during summer. The air temperature inversion occurred frequently in the afternoon during July and August, and sensible heat flux transferred downward (Liu et al., 2011). The wind directions also changed with the height (Fig. 9 in this manuscript). The data showed in figure 11 is a daily averaged value for a whole year; however, the diurnal variations of wind speed, air temperature and relative humidity profile in a day were plotted in figure 10 intending to show the oasis-desert interaction characteristics. Therefore, the profile picture was used in figure 10. Additionally, we also plotted the multi-years mean values (Fig. R1) and found that it could not capture the changing characteristics of wind speed, air temperature and humidity in oasis-desert areas. Therefore, in order to better reflect the oasis-desert interaction characteristics in oasis-desert area, typical days were selected in this manuscript.

[Figure]

Fig.R1 The profile of wind speed, air temperature and relative humidity in January, April, July and October (2012-2021, a: artificial oasis in middle reaches; b: natural oasis in lower reaches)

References:

Liu, S. M., Xu, Z. W., Wang, W. Z., Bai, J., Jia, Z. Z., Zhu, M. J., and Wang, J. M.: A comparison of eddy-covariance and large aperture scintillometer measurements with respect to the energy balance closure problem. Hydrol. Earth Syst. Sci., 15(4), 1291-1306, 2011.

Stoy, P.C., Mauder, M., Foken, T., Marcolla, B., Boegh, E., Ibrom, A., Arain, M., Arneth, A., Aurela, M., Bernhofer, C., Cescatti, A., Dellwik, E., Duce, P., Gianelle, D., Gorsel, E., Kiely, G., Knohl, A., Margolis, H., McCaughey, H., Merbold, L., Montagnanti, L., Papale, D., Reichstein, M., Saunders, M., Serrano-Ortiz, P., Sottocornola, M.,

Spano, D., Vaccari, F., and Varlagin, A: A data-driven analysis of energy balance closure across FLUXNET research sites: The role of landscape scale heterogeneity, Agric. For. Meteorol., 171-172, 137-152, 2013.

Twine, T.E., Kustas, W.P., Norman, J.M., Cook, D.R., Houser, P.R., Meyers, T.P., Prueger, J.H., Starks, P.J., and Wesely, M.L.: Correcting eddy-covariance flux underestimates over a grassland. Agric. For. Meteorol., 103(3), 279-300, 2000.

Xu, Z.W., Ma, Y.F., Liu, S.M., Shi, W.J., and Wang, J.M.: Assessment of the energy balance closure under advective conditions and its impact using remote sensing data, J. Appl. Meteorol. Clim., 56 (1), 127-140, 2017.

Xu, Z. W., Liu, S. M., Zhu, Z. L., Zhou, J., Shi, W. J., Xu, T. R., Yang, X. F., Zhang, Y., and He, X.L.: Exploring evapotranspiration changes in a typical endorheic basin through the integrated observatory network, Agric. For. Meteorol., 290, 108010, 2020.

Zhou, Y., and Li, X.: Energy balance closures in diverse ecosystems of an endorheic river basin, Agric. For. Meteorol., 274, 118-131, 2018.